# Performance of an extended triage questionnaire to detect suspected cases of Severe Acute Respiratory Syndrome Coronavirus 2 (SARS-CoV-2) infection in obstetric patients: Experience from two large teaching hospitals in Lombardy, Northern Italy

Sara Ornaghi[1,2]*, Clelia Callegari[1], Roberta Milazzo[3,4], Laura La Milia[1,2], Federica Brunetti[1,2], Chiara Lubrano[3,4], Chiara Tasca[3,4], Stefania Livio[3,4], Valeria Maria Savasi[3,5], Irene Cetin[3,4], Patrizia Vergani[1,2]

1 Department of Obstetrics and Gynecology, Division of Obstetrics, MBBM foundation at San Gerardo Hospital, Monza, Italy, 2 University of Milan-Bicocca School of Medicine and Surgery, Monza, Italy, 3 Department of Clinical and Biological Sciences, University of Milan, Milan, Italy, 4 Department of Woman, Mother, and Neonate, Buzzi Children's Hospital-ASST Fatebenefratelli-Sacco, Milan, Italy, 5 Department of Woman, Mother, and Neonate, Sacco Hospital-ASST Fatebenefratelli-Sacco, Milan, Italy

* sara.ornaghi@unimib.it

## Abstract

### Objectives

1. To assess the performance of an extended questionnaire in identifying cases of SARS-CoV-2 infection among obstetric patients. 2. To evaluate the rate of infection among health-care workers involved in women's care.

### Study design

A prospective cohort study of obstetric patients admitted to MBBM Foundation and Buzzi Hospital (Lombardy, Northern Italy) from March 16th to May 22nd, 2020. Women were screened on admission by a questionnaire investigating major and minor symptoms of infection and high-risk contacts in the last 14 days. SARS-CoV-2 assessment was performed by RT-PCR on nasopharyngeal swabs. Till April 7th, a targeted SARS-CoV-2 testing triggered by a positive questionnaire was used; from April 8th, a universal testing approach was implemented.

### Results

There were 1,177 women screened by the questionnaire, which yielded a positive result in 130 (11.0%) cases. SARS-CoV-2 RT-PCR was performed in 865 (73.5%) patients, identifying 51 (5.9%) infections. During the first period, there were 29 infected mothers, 4 (13.8%)

**Data Availability Statement:** All relevant data are within the manuscript.

**Funding:** The authors received no specific funding for this work.

**Competing interests:** The authors have declared that no competing interests exist.

**Abbreviations:** SARS-CoV-2, Severe Acute Respiratory Syndrome Coronavirus 2; PPE, personal protective equipment; HCWs, healthcare workers; RT-PCR, reverse transcription-polymerase chain reaction; COVID-19, coronavirus disease.

of whom had a negative questionnaire. After universal testing implementation, there were 22 (3%, 95% CI 1.94% - 4.04%) infected mothers, 13 (59.1%) of whom had a negative questionnaire; rate of infection among asymptomatic women was 1.9%. Six of the 17 SARS-CoV-2-positive women with a negative questionnaire reported symptoms more than 14 but within 30 days before admission. Isolated olfactory or taste disorders were identified in 15.7% of infected patients. Rate of infection among healthcare workers was 5.8%.

## Conclusions

An exhaustive triage questionnaire can effectively discriminate women at low risk of SARS-CoV-2 infection in the context of a targeted and a universal viral testing approach. In 15.7% of infected women, correct classification as a suspected case of infection was due to investigation of olfactory and taste disorders. Extension of the assessed time-frame to 30 days may be worth considering to increase the questionnaire's performance.

## Introduction

The novel Severe Acute Respiratory Syndrome Coronavirus 2 (SARS-CoV-2) has caused a new, unexpected public health emergency. Italy has been particularly affected, with Lombardy being the country epicenter of the SARS-CoV-2 outbreak [1, 2].

Obstetric patients and their caring physicians face unique challenges for their need of in-person visits and hospital admission for delivery.

Identification of suspected cases of SARS-CoV-2 infection at admission is essential for correctly applying isolation measures and use of personal protective equipment (PPE), thus protecting the women, their newborns, and the healthcare workers (HCWs).

Universal screening by reverse transcription-polymerase chain reaction (RT-PCR) of naso-pharyngeal swabs has been proposed as the optimal approach [3–6]. However, limited testing supplies and laboratory workforce may prevent its application in some clinical settings [7]. In addition, although a rapid laboratory testing has been developed [6], most hospitals rely on standard tests with a 5 to 24 hour-turn-around time [8, 9]. This may be a problem when caring for a laboring woman in whom delivery can occur before the RT-PCR result is available.

A targeted screening guided by a structured questionnaire may represent a feasible and valid alternative [5, 10, 11]. Yet, this approach has been questioned due to evidence of high rates of asymptomatic SARS-CoV-2 infection in the obstetric population [4, 6, 12, 13]. However, only major respiratory symptoms were assessed in these studies. Since minor symptoms, such as loss of smell or taste, have been described at earlier stages of infection and also as isolated symptoms in milder forms of the coronavirus disease (COVID-19) [14–20], the possibility of some women being erroneously classified as asymptomatic in these reports has to be considered.

Here we report our data on the use of a comprehensive admission questionnaire for obstetric patients, including both major and minor symptoms of infection as well as high-risk contacts and living environment. Accuracy of the questionnaire in the context of both a targeted and a universal SARS-CoV-2 screening by RT-PCR on nasopharyngeal swabs in two consecutive periods of the outbreak is assessed and discussed herein.

Secondary objective of the study was the assessment of the SARS-CoV-2 infection rate among the HCWs involved in patients' management.

## Material and methods

This was a prospective cohort study of all women admitted to the Obstetric Unit of MBBM Foundation at San Gerardo Hospital and Vittore Buzzi Hospital during pregnancy or the post-partum period from March 16th to May 22nd, 2020.

These hospitals are located in the Milan area, Lombardy region, Northern Italy, and perform approximately 5,600 deliveries per year. Since the beginning of March, strict lockdown measures were in place in this geographic area, which entered a deceleration phase of the outbreak in mid-April [1].

Starting on March 16th, a comprehensive questionnaire including both major and minor symptoms of infection and high-risk contacts in the last 14 days as well as a high-risk living environment (i.e., immigration centers, drug rehabilitation centers) was administered to all women at hospital admission (Fig 1).

The questionnaire was deemed positive when at least one positive answer was present. Support persons (one for each woman) were also screened by means of the questionnaire, and refused hospital access in case of a positive result. Both patients and their support persons were given surgical masks and asked to wear them during their hospital stay; they were also instructed to practice frequent hand sanitization. In addition, all HCWs involved in women's care underwent questionnaire assessment (section A) at the beginning of every shift and wore a surgical mask for the entire shift duration, unless different PPE were required, as means of source control [21, 22].

Initially, a targeted SARS-CoV-2 screening approach triggered by a positive questionnaire and based on RT-PCR testing of nasopharyngeal swabs was used in women with hospital admission after accessing the Emergency Department. In turn, a universal screening was applied to all patients with scheduled admission (i.e., elective pre-labor cesarean section). On April 8th, we changed our policy and started testing all women for SARS-CoV-2 infection independent of the type of hospital admission and the questionnaire result, in agreement with a disposition of the Lombardy Region Health Care Authority.

Cases with scheduled admission underwent RT-PCR testing 24–48 hours in advance in a designated drive-through testing center so results would be available at the time of hospital access to guide isolation measures and use of PPE. Instead, questionnaire results were used for this purpose in cases of unscheduled admission: women with a positive questionnaire were classified as persons under investigation (PUI) and managed accordingly while nucleic acid test results were pending, whereas women with a negative questionnaire were considered not at risk until the result of the RT-PCR test was available.

Nasopharyngeal sampling was performed by a trained resident physician or midwife in appropriate PPE using dedicated swabs. Samples were transferred to the laboratory and processed by RT-PCR testing SARS-CoV-2 with the automated ELITe InGenius system and the GeneFinder COVID-19 Plus RealAmp Kit assay, according to manufacturer's instructions. This assay targets three genes, RNA-dependent RNA polymerase, nucleocapsid protein, and envelope membrane protein, with high specificity. Test results were available in 5 to 24 hours and scored as "positive" or "negative" in both hospitals [9].

Viral testing was also performed in all HCWs involved in patients' management.

The accuracy of the questionnaire to predict SARS-CoV-2 infection in both study periods (March 16th to April 7th and April 8th to May 22nd) was tested by constructing a 2x2 table and calculating sensitivity, specificity, positive predictive value, negative predictive value, positive likelihood ratio (sensitivity/1-specificity), and negative likelihood ratio (1-sensitivity/specificity).

The study was approved by the IRB of the University of Milan-Bicocca and the University of Milan (#15408/2020). A written informed consent was obtained for all women involved in the study.

| | |
|---|---|
| **Name, Last Name** ______________________________ | |
| **Date of Birth** ______________________________ | |

| | |
|---|---|
| **A** | **Symptoms (within the last 14 days)** |
| | Fever ≥ 37.5 °C |
| | Cough |
| | Sore throat |
| | Coryza |
| | Shortness of breath (oxygen saturation < 95% or respiratory rate > 22/min) |
| | Diarrhea |
| | Loss of appetite, nausea, vomiting |
| | Olfactory or taste disorders (loss of smell or taste) |
| | Myalgias/arthralgias |
| | Severe headache |
| | Conjunctivitis |
| **B** | **History of high-risk contacts (within the last 14 days) or high-risk living environment** |
| | High-risk occupation (healthcare worker, laboratory technician) |
| | Contact with a known or suspected COVID-19 case |
| | High-risk living environment (immigration center, rehabilitation center) |
| | *Total positive items section A + B* |
| **Date** ________________________________________________ | |
| **Healthcare worker's name and signature** ________________________________ | |
| **Patient's signature** ________________________________________ | |

**Fig 1. Admission questionnaire.** Adapted from Poon *et al.* [11].

## Results

A total of 1,177 women were assessed at hospital admission by the questionnaire during the study period (n = 447 at MBBM Foundation at San Gerardo Hospital and n = 730 at Vittore Buzzi Hospital). Nine-hundred and forty-five (80.3%) women were admitted to the L&D unit, whereas 196 (16.7%) and 36 (3.0%) women to the antepartum and postpartum unit, respectively.

Of the 1,177 patients assessed, 865 (73.5%) were tested for SARS-CoV-2 by RT-PCR on nasopharyngeal swab and 51 (5.9%) were positive.

Between March 16th and April 7th, RT-PCR testing was performed in 129 out of 441 patients. Questionnaire was positive on admission in 63 (14.3%) women. Among the 319 patients with unscheduled admission and a negative questionnaire, 7 (2.2%) were tested for SARS-CoV-2 during hospitalization because of onset of fever. All of them were negative.

SARS-CoV-2 infection was diagnosed in 29 mothers, 4 (13.8%) of whom had a negative questionnaire. One of these 4 patients failed to report high-risk contacts (i.e., fever and cough in close family members a few days before admission). An additional 2 women revealed symptoms suggestive of SARS-CoV-2 infection occurring more than 2 weeks but within one month before admission. None of the 4 patients developed symptoms during hospitalization.

After implementing universal viral screening, we identified 67/736 positive questionnaires. Nasopharyngeal swab analysis by RT-PCR recognized 22 (3%, 95% CI 1.94% - 4.04%) cases of SARS-CoV-2 infection. Questionnaire was negative in 13 (59.1%) of them, for a rate of infection among asymptomatic women of 1.9%. Four out of these 13 women reported loss of taste or smell more than 14 days but within one month before admission. None of the 13 patients developed symptoms during hospitalization.

Accuracy of the questionnaire to predict SARS-CoV-2 infection in both study periods is shown in Table 1.

Detailed assessment of positive questionnaires showed that fever $\geq 37.5°C$ was the most common positive item (42.3%), followed by high-risk contacts/living environment (30.8%), cough (25.4%), gastrointestinal symptoms (13.8%), and loss of smell or taste (11.5%) (Fig 2A).

Fever and cough were more commonly identified during the first study period compared to the second one (57.1% *versus* 23.4% and 30.2% *versus* 20.9%, respectively), whereas gastrointestinal symptoms displayed an opposite trend (7.9% *versus* 19.4%). Rate of olfactory and taste disorders, as well as of high-risk contacts/living environment, remained stable over time.

Of the fifteen women who reported loss of smell or taste within 14 days before admission, this was the only positive questionnaire item in three. Also, when the time-frame of investigation was extended to the last 30 days before admission, an additional five mothers were identified with isolated olfactory or taste disorders. All these eight patients tested positive for SARS-CoV-2.

Frequency and time-trend of positive questionnaire items among SARS-CoV-2 infected women is shown in Fig 2B.

There were 307 HCWs involved in patients' management in both hospitals during the overall study period. Eighteen of them (5.8%) tested positive for SARS-CoV-2. There were no cases of moderate or severe COVID-19.

## Discussion

Our study investigated the accuracy of a comprehensive questionnaire thoroughly assessing obstetric patients upon hospital admission to identify cases suspected for SARS-CoV-2 infection.

**Table 1. Accuracy of the admission questionnaire in the two study periods.**

| First study period (March 16[th]–April 7[th], 2020) | | | |
|---|---|---|---|
| - Targeted SARS-CoV-2 screening - | | | |
| | Positive RT-PCR for SARS-CoV-2 | Negative RT-PCR for SARS-CoV-2 | Total |
| Positive questionnaire | 25 | 38 | 63 |
| Negative questionnaire | 4 * | 55 | 59 |
| Total | 29 | 93 | 122 |

- Sensitivity = 25/29, 86.2%; specificity = 55/93, 59.1%
- Positive predictive value = 25/63, 39.7%; negative predictive value = 55/59, 93.2%
- Positive likelihood ratio = 2.11; negative likelihood ratio = 0.23

| Second study period (April 8[th]–May 22[nd], 2020) | | | |
|---|---|---|---|
| - Universal SARS-CoV-2 screening - | | | |
| | Positive RT-PCR for SARS-CoV-2 | Negative RT-PCR for SARS-CoV-2 | Total |
| Positive questionnaire | 9 | 58 | 67 |
| Negative questionnaire | 13 # | 656 | 669 |
| Total | 22 | 714 | 736 |

- Sensitivity = 9/22, 40.9%; specificity = 656/714, 91.9%
- Positive predictive value = 9/67, 13.4%; negative predictive value = 656/669, 98.1%
- Positive likelihood ratio = 5.13; negative likelihood ratio = 0.64

* One patient failed to report exposure to high-risk contacts a few days before admission, and two patients reported fever and dry cough (n = 1) and loss of smell and taste (n = 1) more than 14 days but within 30 days before admission.

# Four patients reported loss of smell or taste more than 14 days but within 30 days before admission.

Differently from previous reports [4, 6–8, 13], our questionnaire evaluated the presence of not only major respiratory symptoms of SARS-CoV-2 infection, including fever, cough, and shortness of breath, but also minor symptoms, such as loss of smell or taste, as well as high-risk contacts during the last two weeks preceding admission.

We observed a negative predictive value for SARS-CoV-2 infection of 93.2% and 98.1% in the context of a targeted and a universal viral screening, respectively, in two consecutive periods of the outbreak. In addition, we identified a low rate of viral infection among the HCWs involved in women's care [23–25].

The ability of accurately discriminating women at low risk for infection is pivotal in case of both a targeted and a universal SARS-CoV-2 screening. In clinical settings with no ability to perform a universal testing, a well-performing admission questionnaire can adequately guide a targeted screening approach and still allow protection of the patients and the HCWs. On the other hand, when a universal screening approach is feasible, the questionnaire allows appropriate patients' cohorting and application of isolation measures and contact precautions while RT-PCR results are pending. This is particularly important to prevent potential viral spread to other patients and HCWs and mother-to-child transmission when caring for laboring women, in whom delivery can occur before nucleic acid test results are available. In fact, turn-around time of RT-PCR testing is usually >5 hours in most facilities [8, 9] and availability of rapid testing is limited [6].

Much of the push for implementation of a universal screening with SARS-CoV-2 RT-PCR hinges on avoiding unintended exposures to HCWs when caring for an asymptomatic patient, especially in geographic areas with a high community prevalence of COVID-19 [26, 27]. Our hospitals are located in the epicenter of SARS-CoV-2 outbreak in Italy [1, 2]. Nonetheless, we had only 1.9% asymptomatic, SARS-CoV-2 positive women during the universal screening period. This rate is 7 times lower than that reported by other centers in similarly hardly hit

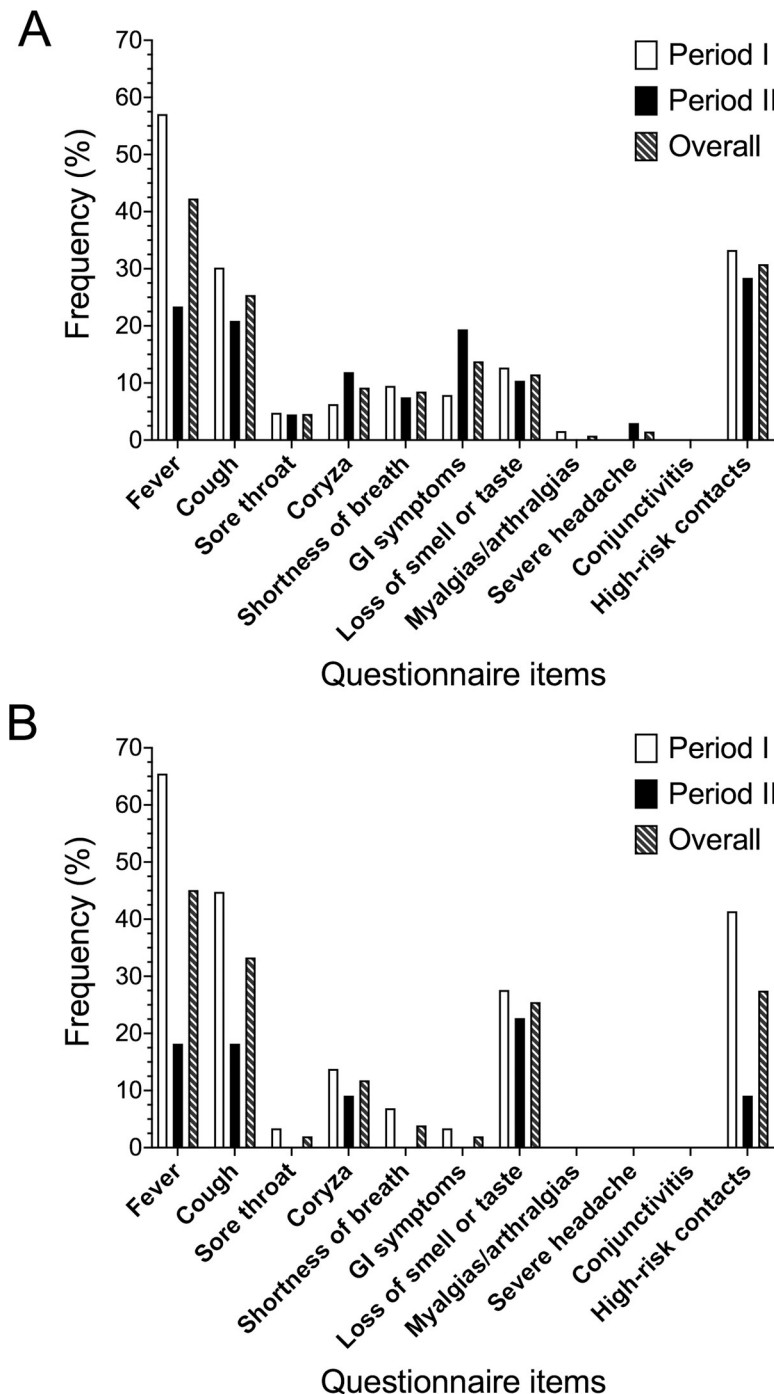

**Fig 2.** Distribution of positive questionnaire items among women with a positive questionnaire on admission (A) and with a positive SARS-CoV-2 RT-PCR testing (B) during the study period. Fever was considered for values ≥37.5˚C; shortness of breath was defined as oxygen saturation <95% or respiratory rate >22/min; GI (gastrointestinal) symptoms included loss of appetite, nausea, vomiting, and diarrhea; high-risk contacts refer to high-risk occupation, contact with a known or suspected COVID-19 case and high-risk living environment. Period I: from March 16th to April 7th, 2020; Period II: from April 8th to May 22nd, 2020.

areas, such as New York City [4, 6]. A different phase of the outbreak during the study period may have contributed to this difference [1, 28]. However, our detailed investigation of minor

symptoms, such as loss of smell or taste, may also have played an important role [3, 29]. Olfactory and taste disorders have been reported as the only symptoms of infection in up to 17% of SARS-CoV-2 positive individuals, and to display a higher predictive ability of having the virus than fever or persistent cough [16, 18, 19]. Isolated loss of smell or taste was identified in 15.7% of our virally infected patients. In addition, frequency of olfactory and taste disorders did not differ between the two study periods, whereas that of fever and cough displayed a substantial reduction (Fig 2A and 2B). These data suggest that patients' assessment upon admission during an outbreak by a novel air-tract pathogen should include not only major respiratory but also minor symptoms, especially in the more advanced phases of the outbreak.

Prolonged SARS-CoV-2 RT-PCR positivity at more than 2 weeks from symptom onset has been reported in infected individuals [30, 31]. Overall, we identified 51 women with SARS-CoV-2 infection, and in 17 (33.3%) questionnaire upon admission was deemed negative. However, 6 (35.3%) of these patients reported symptoms suggestive of infection more than 14 but within 30 days before hospital admission. Had the questionnaire investigated a one-month time-frame, negative predictive values would have increased to 98.2% and 98.6% in the targeted and the universal viral screening period, respectively. Of note, whether such patients would have been infectious and thus able to spread the virus at the time of admission is still a matter of debate [32–36]. Similarly, the infectiousness of fully asymptomatic women with positive SARS-CoV-2 RT-PCR results is unclear [31, 33, 36–38]. Unfortunately, we could not address this issue in our patients since we did not perform viral culture experiments [33, 34]. Nonetheless, independent of the infectiousness potential of these women, widespread use of face masks and frequent hand sanitization have likely contributed to successful viral spread control in our units [21, 22].

Strengths of our study are the following. First, it was conducted in two large teaching hospitals in the Italian epicenter of the outbreak, thus providing useful data for equally affected areas. Second, it investigated the questionnaire's performance in the context of both a targeted and a universal viral screening approach in two consecutive periods of the outbreak. Third, it assessed the universal screening approach over a 6-week time period, which may have allowed to better capture the real trend of infection over time among obstetric patients than much shorter study periods [4, 6, 8].

Admission questionnaires may have limitations since they rely on honest answering. The possibility of patients being not completely sincere due to fear of isolation measures, especially in laboring women, has to be considered. This event occurred in at least one woman in our cohort. Addition of objective, point-of-care parameters, such as lymphocyte count and lung ultrasound, to the admission screening procedure to increase its accuracy may be worth exploring [7, 10, 27, 31, 39–41].

Another limitation of our study is that SARS-CoV-2 testing by RT-PCR on nasopharyngeal swabs was performed in a targeted manner during the first study period, thus leading to 312 untested women. This also prevented a meaningful comparison of infection rates between the two study periods with a targeted and a universal SARS-CoV-2 testing approach, respectively.

## Conclusions

With recognition that a "one-size-fits-all" approach is unlikely to be justifiable [26], decisions regarding universal viral testing should be made in the context of regional prevalence of SARS-CoV-2 infection as well as financial and human resources and PPE availability in each obstetric unit.

Our data show that thorough assessment of obstetric patients upon hospital admission by means of an exhaustive questionnaire is feasible and effective in discriminating women at low

risk of SARS-CoV-2 infection in the context of both a targeted and a universal screening approach. Extension of the investigated time-frame from 14 to 30 days may be worth considering to increase the questionnaire's performance, especially in this high-risk population. The question remains whether this group of women, as well as of those SARS-CoV-2 positive but fully asymptomatic, represents an actual source of viral spread.

## Author Contributions

**Conceptualization:** Sara Ornaghi, Irene Cetin, Patrizia Vergani.

**Data curation:** Sara Ornaghi, Clelia Callegari, Roberta Milazzo, Laura La Milia, Federica Brunetti, Chiara Lubrano, Chiara Tasca, Stefania Livio.

**Formal analysis:** Sara Ornaghi, Clelia Callegari, Roberta Milazzo, Laura La Milia, Federica Brunetti, Chiara Lubrano, Chiara Tasca, Stefania Livio.

**Methodology:** Sara Ornaghi.

**Project administration:** Patrizia Vergani.

**Supervision:** Sara Ornaghi, Valeria Maria Savasi, Irene Cetin.

**Writing – original draft:** Sara Ornaghi.

**Writing – review & editing:** Sara Ornaghi, Valeria Maria Savasi, Irene Cetin, Patrizia Vergani.

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
