## [Decision Letter · Decision Letter 0]

18 Aug 2020

PONE-D-20-23463

Performance of an extended triage questionnaire to detect suspected cases of Severe Acute Respiratory Syndrome Coronavirus 2 (SARS-CoV-2) infection in obstetric patients: experience from two large teaching hospitals in Lombardy, Northern Italy.

PLOS ONE

Dear Dr.ssa Ornaghi,

Thank you for submitting your manuscript to PLOS ONE. After careful consideration, we feel that it has merit but does not fully meet PLOS ONE’s publication criteria as it currently stands. Therefore, we invite you to submit a revised version of the manuscript that addresses the points raised during the review process.

We look forward to receiving your revised manuscript.

Kind regards,

Francesco Di Gennaro

Academic Editor

PLOS ONE

Additional Editor Comments:

Dear Authors,

I appreciate a lot your manuscript.

Follow the reviewer suggestions to improve your article.

Journal Requirements:

Reviewers' comments:

Reviewer's Responses to Questions

**Comments to the Author**

1. Is the manuscript technically sound, and do the data support the conclusions?

Reviewer #1: Yes

Reviewer #2: Yes

2. Has the statistical analysis been performed appropriately and rigorously? 

Reviewer #1: No

Reviewer #2: Yes

3. Have the authors made all data underlying the findings in their manuscript fully available?

Reviewer #1: Yes

Reviewer #2: Yes

4. Is the manuscript presented in an intelligible fashion and written in standard English?

Reviewer #1: Yes

Reviewer #2: Yes

5. Review Comments to the Author

Reviewer #1: The authors sought to assess the performance of an extended questionnaire in identifying cases of SARS COV-2 infection among obstetric patients.

This study was conducted in two of the biggest birth centers in the epicenter of pandemic in Italy (Lombardy).

They individuated two study periods:

- in the first period the RT-PCR test on nasopharyngeal swabs was targeted on the base of a positive questionnaire.

- in the second study period, a universal approach was applied.

- in both study periods the questionnaire was administered to every patients

I found the argument of particular interest because I think it's laudable the effort of trying to include also minor symptoms, that were not initially included in the official questionnaire proposed by the main institution (ISUOG and regional normative in Lombardy).

The questionnaire is also a good alternative for those contexts in which a universal RT-PCR approach is not feasible for limited resources.

This is the reason why I ask for Minor Revision to this paper.

Here are my suggestions:

Abstract

- results, page 2 line 36-41:

I will rewrite in this way: after universal testing implementation, there were 22 infected mothers, 13 (59%) of them had a negative questionnaire.

Methods

I propose to perform ROC curve to assess the performance of the questionnaire in the "universal approach" period.

Discussion

I would add in the discussion (page 14 lines 269-274) that one limit of the study is that the RT-PCR on nasopharingeal swabs was not performed in all the women that fulfilled the questionnaire (865/1,177).

Reviewer #2: August 2020

Review- Performance of an extended triage questionnaire to detect suspected cases of severe acute respiratory syndrome Corovavirus 2 infection obstetric patients: experience from two large teaching hospitals in Lombardy, Northern Italy

Thank you for the opportunity to review this manuscript.

This study explores the plausibility of a questionnaire in the screening of obstetric patients for Covid.

Since the beginning of 2020 Covid has brought turmoil over the entire globe and health care systems are seeking ways to confront many issues that came with it, both in patient care and in safe guarding health care workers.

Abstract

Should be revised with a native speaker.

Introduction

Well written.

Methods

Well written.

Results

Well written, clearly understood.

Discussion

Well written, I would add a section of limitations. The author do write regarding the problem of bias in the answers, but do not address the different rates of positive swabs- 29/122 in the first time period versus 22/736 in the second. There are different explanations for this, the first is the increase in swabs preformed- meaning that the 22 positive were just the tip of the iceberg and the questionnaire has a larger false negative than expected. The second is a change in the infection rate in general. Please address these points.

6. PLOS authors have the option to publish the peer review history of their article (what does this mean?). If published, this will include your full peer review and any attached files.

Reviewer #1: **Yes: **Annalisa Inversetti

Reviewer #2: **Yes: **Yael Baumfeld

---

## [Author Response · Author response to Decision Letter 0]

27 Aug 2020

Dear Referees:

Thank you for the strong positive comments on the first submission of our manuscript. We also appreciate the helpful suggestions as to how we might further improve the paper. Below we respond to each suggestion with details about how we amended the manuscript.

Response to Referees.

Response to Referee 1. 

We appreciate the comments from Referee 1 suggesting our study was of particular interest due to inclusion of minor symptoms in the proposed screening questionnaire. We agree this is extremely important to improve screening efficiency for SARS-CoV-2 infection among obstetric patients at hospital admission. 

The Referee asked to rewrite a sentence of the Abstract reporting results of the universal screening period. We addressed this request (Abstract section, Line 36-38). 

Referee 1 proposes to perform a ROC curve analysis to assess the questionnaire performance during the study period when a universal SARS-CoV-2 screening approach was implemented. 

We thank the reviewer for this comment; however, since the proposed screening questionnaire has a dichotomous result (positive/negative), the most appropriate way of evaluating its accuracy is by calculation of sensitivity, specificity, and positive and negative predictive values in a two-by-two table (as reported in Table 1, Line 182). In turn, a ROC curve analysis with calculation of the area under the curve (AUC) is a more effective measure of assessing a test accuracy when the test generates ordinal or continuous results; sensitivity and specificity can be then computed across all the possible threshold values and vary across the different thresholds [1, 2].

The Referee suggests to specify that a limitation of our study is that RT-PCR on nasopharyngeal swabs for diagnosing SARS-CoV-2 infection was not performed in all women screened by the questionnaire. We appreciate this comment and it has now been addressed in the Discussion section, Line 280-282. 

Response to Referee 2. 

We thank the Referee for the positive comments on our manuscript.

The abstract has now been revised. 

Referee 2 suggests to add a section on limitations of our work. This has been done in the Discussion section, Line 280-283.

References

1. Søreide K. Receiver-operating characteristic curve analysis in diagnostic, prognostic and predictive biomarker research. Journal of clinical pathology. 2009;62(1):1-5. Epub 2008/09/27. doi: 10.1136/jcp.2008.061010. PubMed PMID: 18818262.

2. Hulley S, Cummings S, Browner W, Grady D, Newman T. Designing Clinical Research. Fourth ed: Lippincott Williams & Wilkins, Wolters Kluwer; 2013.

---

## [Decision Letter · Decision Letter 1]

2 Sep 2020

Performance of an extended triage questionnaire to detect suspected cases of Severe Acute Respiratory Syndrome Coronavirus 2 (SARS-CoV-2) infection in obstetric patients: experience from two large teaching hospitals in Lombardy, Northern Italy.

PONE-D-20-23463R1

Dear Dr.ssa Sara Ornaghi,

We’re pleased to inform you that your manuscript has been judged scientifically suitable for publication and will be formally accepted for publication once it meets all outstanding technical requirements.

Kind regards,

Francesco Di Gennaro

Academic Editor

PLOS ONE

Additional Editor Comments (optional):

Dear Authors,

congratulations for your manuscript that now can be accept!

Reviewers' comments:

Reviewer's Responses to Questions

**Comments to the Author**

1. If the authors have adequately addressed your comments raised in a previous round of review and you feel that this manuscript is now acceptable for publication, you may indicate that here to bypass the “Comments to the Author” section, enter your conflict of interest statement in the “Confidential to Editor” section, and submit your "Accept" recommendation.

Reviewer #1: All comments have been addressed

Reviewer #2: All comments have been addressed

2. Is the manuscript technically sound, and do the data support the conclusions?

Reviewer #1: Yes

Reviewer #2: Yes

3. Has the statistical analysis been performed appropriately and rigorously? 

Reviewer #1: Yes

Reviewer #2: Yes

4. Have the authors made all data underlying the findings in their manuscript fully available?

Reviewer #1: (No Response)

Reviewer #2: Yes

5. Is the manuscript presented in an intelligible fashion and written in standard English?

Reviewer #1: Yes

Reviewer #2: Yes

6. Review Comments to the Author

Reviewer #1: (No Response)

Reviewer #2: The manuscript is interesting and all comments have been answered fully.

I recommend accepting it for publication

7. PLOS authors have the option to publish the peer review history of their article (what does this mean?). If published, this will include your full peer review and any attached files.

Reviewer #1: **Yes: **Annalisa Inversetti

Reviewer #2: **Yes: **Yael Baumfeld

---

## [Editor Report · Acceptance letter]

7 Sep 2020

PONE-D-20-23463R1 

Performance of an extended triage questionnaire to detect suspected cases of Severe Acute Respiratory Syndrome Coronavirus 2 (SARS-CoV-2) infection in obstetric patients: experience from two large teaching hospitals in Lombardy, Northern Italy. 

Dear Dr. Ornaghi:

I'm pleased to inform you that your manuscript has been deemed suitable for publication in PLOS ONE. Congratulations! Your manuscript is now with our production department. 

Kind regards, 

on behalf of

Dr. Francesco Di Gennaro 

Academic Editor

PLOS ONE